# Attending to Graph Transformers

**Luis Müller**                                   *luis.mueller@cs.rwth-aachen.de*
*Department of Computer Science*
*RWTH Aachen University*

**Mikhail Galkin**                                   *mikhail.galkin@intel.com*
*Intel AI Lab*

**Ladislav Rampášek**                              *rampasek@isomorphiclabs.com*
*Isomorphic Labs*

**Christopher Morris**                             *morris@cs.rwth-aachen.de*
*Department of Computer Science*
*RWTH Aachen University*

**Reviewed on OpenReview:** *https: // openreview. net/ forum? id= HhbqHBBrfZ*

## Abstract

Recently, transformer architectures for graphs emerged as an alternative to established techniques for machine learning with graphs, such as (message-passing) graph neural networks. So far, they have shown promising empirical results, e.g., on molecular prediction datasets, often attributed to their ability to circumvent graph neural networks' shortcomings, such as over-smoothing and over-squashing. Here, we derive a taxonomy of graph transformer architectures, bringing some order to this emerging field. We overview their theoretical properties, survey structural and positional encodings, and discuss extensions for important graph classes, e.g., 3D molecular graphs. Empirically, we probe how well graph transformers can recover various graph properties, how well they can deal with heterophilic graphs, and to what extent they prevent over-squashing. Further, we outline open challenges and research direction to stimulate future work. Our code is available at `https://github.com/luis-mueller/probing-graph-transformers`.

## 1 Introduction

Graph-structured data are prevalent across application domains ranging from chemo- and bioinformatics (Barabasi & Oltvai, 2004; Reiser et al., 2022) to image (Simonovsky & Komodakis, 2017) and social-network analysis (Easley & Kleinberg, 2010), underlining the importance of machine learning methods for graph data. In recent years, *(message-passing) graph neural networks* (GNNs) (Chami et al., 2022; Gilmer et al., 2017; Morris et al., 2021) were the dominant paradigm in machine learning for graphs. However, with the rise of transformer architectures (Vaswani et al., 2017) in natural language processing (Lin et al., 2021b) and computer vision (Han et al., 2022), recently, a large number of works in the field focused on designing transformer architectures capable of dealing with graphs, so-called *graph transformers* (GTs).

Graph transformers have already shown promising performance (Ying et al., 2021), e.g., by topping the leaderboard of the OGB Large-Scale Challenge (Hu et al., 2021; Masters et al., 2022) in the molecular property prediction track. The superiority of GTs over standard GNN architecture is often explained by GNNs' bias towards encoding local structure and being unable to capture global or long-range information, often attributed to phenomena such as *over-smoothing* (Li et al., 2018), *under-reaching* (Barceló et al., 2020), and *over-squashing* (Alon & Yahav, 2021; Di Giovanni et al., 2023). Many papers (Rampášek et al., 2022) speculate that GTs do not suffer from such effects as they aggregate information over all nodes in a given graph and hence are not limited to local structure bias. However, to make GTs aware of graph structure, one has to equip them with so-called *structural* and *positional*

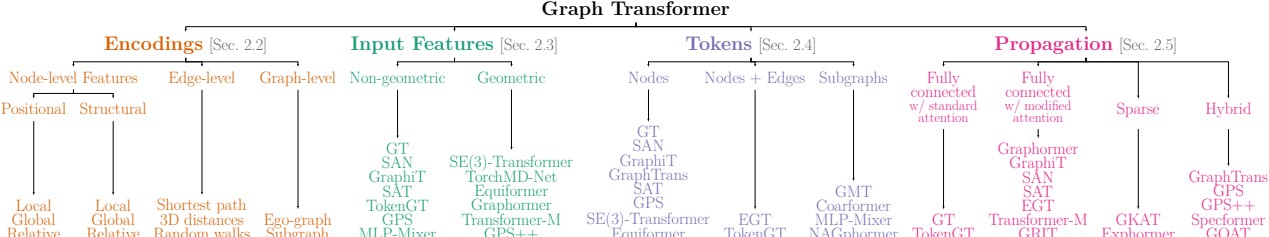

Figure 1: Categorization of graph transformers along four main categories with representative architectures: Encodings; see Section 2.2, input features; see Section 2.3, tokens; see Section 2.4, propagation; see Section 2.5. See also Figure 3 detailing how the different branches translate into changes to the original transformer.

*encodings.* Here, structural encodings are, e.g., additional node features to make the GT aware of (sub-)graph structure. In contrast, positional encodings make a node aware of its position in the graph concerning the other nodes.

**Present Work.** Here, we derive a taxonomy of state-of-the-art GT architectures, giving an organized overview of recent developments. In addition, we survey common positional and structural encodings and clarify how they are related to GTs' theoretical properties, e.g., their expressive power to capture graph structure. Additionally, we investigate these properties empirically by probing how well GTs can recover various graph properties, deal with heterophilic graphs, and to what extent GTs alleviate the over-squashing phenomenon. Further, we outline open challenges and research directions to stimulate future work. Our categorization, theoretical clarification, and experimental study present a useful handbook for the GT and the broader graph machine-learning community. We hope that its insights and principles will help spur novel research results and avenues.

**Related Work.** Since GTs emerged recently, only a few surveys exist. Notably, Min et al. (2022a) provide a high-level overview of some of the recent GT architectures. Different from the present work, they do not discuss GTs' theoretical and practical shortcomings and miss out on recent architectural advancements. Chen et al. (2022a) gives an overview of GTs for computer vision. Finally, Rampášek et al. (2022) provide a general recipe for classifying GT architectures, focusing on devising empirically well-performing architectures rather than giving a detailed, principled overview of the literature.

## 1.1 Background

A *graph* $G$ is a pair $(V(G), E(G))$ with a *finite* set of *nodes* $V(G)$ and a set of *edges* $E(G) \subseteq \{\{u, v\} \subseteq V \mid u \neq v\}$. For ease of notation, we denote an edge $\{u, v\}$ as $(u, v)$ or $(v, u)$. In the case of *directed graphs*, $E(G) \subseteq \{(u, v) \in V(G)^2 \mid u \neq v\}$. Throughout the paper, we set $n := |V(G)|$ and $m := |E(G)|$. A *node-attributed graph* $G$ is a triple $(V(G), E(G), \mathbf{X})$, where $\mathbf{X} \in \mathbb{R}^{n \times d}$, for $d > 0$, is a *node feature matrix* and $\mathbf{X}_v$ is the node feature of node $v \in V(G)$. Similarly, we can represent edge features by an *edge feature matrix* $\mathbf{E} \in \mathbb{R}^{m \times e}$, for $e > 0$, where $\mathbf{E}_{vw}$ is the edge feature of edge $(v, w) \in E(G)$. The *neighborhood* of $v \in V(G)$ is $N(v) := \{u \in V(G) \mid (v, u) \in E(G)\}$. We say that two graphs $G$ and $H$ are *isomorphic* if there exists an edge-preserving bijection $\varphi \colon V(G) \to V(H)$, i.e., $(u, v)$ is in $E(G)$ if and only if $(\varphi(u), \varphi(v))$ is in $E(H)$ for all $u, v \in V(G)$. We denote a multiset by $\{\!\{\ldots\}\!\}$.

**Equivariance and Invariance.** Ideally, machine learning models for graphs respect their symmetries, such as being agnostic to the node permutations or other (group) transformations, such as rotation, leading to the definitions of *equivariance* and *invariance*. In particular, we can expect better generalization for models respecting these symmetries (Petrache & Trivedi, 2023). In general (Fuchs et al., 2021), given a transformation $\mathbf{T}$, a function $f$ is equivariant if transforming the vector input $\mathbf{x}$ is equal to transforming the output of the function $f$, i.e., $f(\mathbf{T}\mathbf{x}) = \mathbf{T}f(\mathbf{x})$. A function $g$ is invariant if transforming the vector input $\mathbf{x}$ does not change the output, i.e., $g(\mathbf{T}\mathbf{x}) = g(\mathbf{x})$. In the 3D Euclidean space, 3D translations, rotations, and reflections form the $E(3)$ group. Translation and rotation form the $SE(3)$ group. Rotations form the $SO(3)$ group, rotations and reflections form the $O(3)$ group.

**Graph Transformers.** A transformer is a stack of alternating blocks of *multi-head attention* and fully-connected *feed-forward networks*. Let $G$ be a graph with node feature matrix $\mathbf{X} \in \mathbb{R}^{n \times d}$.[1] In each layer, $t > 0$, given node feature matrix $\mathbf{X}^{(t)} \in \mathbb{R}^{n \times d}$, a single attention head computes

$$\mathsf{Attn}(\mathbf{X}^{(t)}) := \mathrm{softmax}\left(\frac{\mathbf{Q}\mathbf{K}^T}{\sqrt{d_k}}\right)\mathbf{V}, \tag{1}$$

where the softmax is applied row-wise, $d_k$ denotes the feature dimension of the matrices $\mathbf{Q}$ and $\mathbf{K}$, with $\mathbf{X}^{(0)} := \mathbf{X}$. Here, the matrices $\mathbf{Q}, \mathbf{K}$, and $\mathbf{V}$ are the result of projecting $\mathbf{X}^{(t)}$ linearly,

$$\mathbf{Q} := \mathbf{X}^{(t)}\mathbf{W}_Q, \quad \mathbf{K} := \mathbf{X}^{(t)}\mathbf{W}_K, \quad \text{and} \quad \mathbf{V} := \mathbf{X}^{(t)}\mathbf{W}_V,$$

using three matrices $\mathbf{W}_Q, \mathbf{W}_K \in \mathbb{R}^{d \times d_K}$, and $\mathbf{W}_V \in \mathbb{R}^{d \times d}$, with optional bias terms omitted for clarity. Now, multi-head attention $\mathsf{MultiHead}(\mathbf{X}^{(t)})$ concatenates multiple (single) attention heads, followed by an output projection to the feature space of $\mathbf{X}^{(t)}$. By combining the above with additional residual connections and normalization, the transformer layer updates features $\mathbf{X}^{(t)}$ via

$$\mathbf{X}^{(t+1)} := \mathsf{FFN}\big(\mathsf{MultiHead}(\mathbf{X}^{(t)}) + \mathbf{X}^{(t)}\big). \tag{2}$$

**Graph Neural Networks.** Intuitively, GNNs learn a vector representing each node in a graph by aggregating information from neighboring nodes. Formally, let $G$ be a graph with node feature matrix $\mathbf{X} \in \mathbb{R}^{n \times d}$. A GNN architecture consists of a stack of neural network layers, i.e., a composition of permutation-equivariant parameterized functions. Each layer aggregates local neighborhood information, i.e., the neighbors' features, around each node and then passes this aggregated information on to the next layer. Following Gilmer et al. (2017) and Scarselli et al. (2009), in each layer, $t \geq 0$, we compute node features

$$\mathbf{h}_v^{(t+1)} := \mathsf{UPD}^{(t)}\Big(\mathbf{h}_v^{(t)},\ \mathsf{AGG}^{(t)}\big(\{\!\!\{\mathbf{h}_u^{(t)} \mid u \in N(v)\}\!\!\}\big)\Big) \in \mathbb{R}^d,$$

where $\mathsf{UPD}^{(t)}$ and $\mathsf{AGG}^{(t)}$ may be differentiable parameterized functions, e.g., neural networks. For example, GNNs often compute a vector for node $v$ by using sum aggregation (Morris et al., 2019), i.e.,

$$\mathbf{h}_v^{(t+1)} := \sigma\Big(\mathbf{h}_v^{(t)}\mathbf{W}_1^{(t)} + \sum_{w \in N(v)} \mathbf{h}_w^{(t)}\mathbf{W}_2^{(t)}\Big),$$

where $\sigma$ is a non-linearity applied in a point-wise fashion, $\mathbf{W}_1^{(t)}$ and $\mathbf{W}_2^{(t)} \in \mathbb{R}^{d \times d}$ are parameter matrices, and $\mathbf{h}_v^{(0)} := \mathbf{X}_v$. As noticed by Mialon et al. (2021), we can rewrite attention as defined in Eq. (1) as

$$\mathsf{Attn}(\mathbf{X}^{(t)})_v = \sum_{u \in V(G)} \frac{k_{\exp}(\mathbf{X}_v^{(t)}, \mathbf{X}_u^{(t)})}{\sum_{w \in V(G)} k_{\exp}(\mathbf{X}_v^{(t)}, \mathbf{X}_w^{(t)})}\mathbf{X}_u^{(t)}\mathbf{W}_V, \tag{3}$$

for $v \in V(G)$, where

$$k_{\exp}(\mathbf{X}_v^{(t)}, \mathbf{X}_w^{(t)}) := \exp\big(\mathbf{X}_v^{(t)}\mathbf{W}_Q \mathbf{X}_w^{(t)}\mathbf{W}_K / \sqrt{d_K}\big)$$

and obtain an aggregation reminiscent of a GNN. In fact, we may view a GT as a special GNN, operating on a complete graph, where the attention score weights the importance of each node during the sum aggregation. Conversely, Veličković et al. (2018) propose graph attention networks (GAT), a GNN that replaces sum aggregation with self-attention. We note that their definition of self-attention differs from Equation (1) in that GAT replaces the dot-product with concatenation and subsequent linear projection.

## 2 The Landscape of Graph Transformers

In the following, we outline our taxonomy of GTs, see also Figure 1, bringing some order to the growing set of GT architectures. We start by discussing the theoretical properties of GTs that heavily rely on structural and

---

[1]For simplicity, we learn attention between a graph's nodes. However, in Section 2.4, we extend this to, e.g., edges or subgraphs.

positional encodings, which we study subsequently. Further, we discuss different approaches to dealing with essential classes of input node features, e.g., 3D coordinates in the case of molecules. We then study how to *tokenize* a graph, i.e., partition a graph into atomic entities between which the attention is computed, e.g., nodes. Then, we review how GTs organize message propagation in the graph through global, sparse, or hybrid attention. Finally, we overview representative applications of GTs.

## 2.1 Theoretical Properties

It is crucial to understand that the general GT architecture of Eq. (2) is less expressive in distinguishing non-isomorphic graphs than standard GNNs. Hence, it is also weaker in approximating permutation-invariant and -equivariant functions over graphs (Chen et al., 2019). GTs are weaker since, without sufficiently expressive structural and positional encodings, they cannot capture any graph structure besides the number of nodes and hence equal DeepSets-like architectures (Zaheer et al., 2020) in expressive power. Thus, for GTs to capture non-trivial graph structure information, they are crucially dependent on such encodings; see below. In fact, by leveraging the results in Chen et al. (2019), it is easy to show that GTs can become maximally expressive, i.e., universal function approximators, if they have access to maximally expressive structural bias, e.g., structural encodings. However, this is equivalent to solving the graph isomorphism problem (Chen et al., 2019). Moreover, we stress that GNN architectures equipped with the same encodings will also possess the same expressive power. Hence, regarding expressive power, GTs do not have an advantage over GNNs.

Cai et al. (2023) study the connection between GNNs and graph transformers, showing that under mild assumptions, GNNs with a virtual node connected to all other nodes are sufficient to simulate a graph transformer. Conversely, Kim et al. (2022) propose a hierarchy of (higher-order) transformers and respective positional encodings that is aligned with $k$-IGNs (Maron et al., 2019), i.e., for each $k > 1$, there exists a corresponding higher-order transformer that can simulate a $k$-IGN. The relationship between $k$-IGNs and $k$-WL (Maron et al., 2019) then implies the connection between transformers and the $k$-WL hierarchy as well.

## 2.2 Structural and Positional Encodings

As outlined in the previous subsection, GTs are crucially dependent on structural and positional encodings to capture graph structure. Although there is no formal definition or distinction between the two, structural encodings make the GT aware of graph structure on a *local*, *relative*, or *global* level. Such encodings can be attached to node-, edge-, or graph-level features. Examples of local structural encodings include annotating node features with node degrees (Chen et al., 2022b), the diagonal of the $m$-step random-walk matrix (Dwivedi et al., 2022), the time-derivative of the heat-kernel diagonal (Kreuzer et al., 2021), enumerate or count predefined substructures and the node's role within (Bouritsas et al., 2022), or Ricci curvature (Topping et al., 2022). Examples of edge-level relative structural encodings include relative shortest-path distances (Chen et al., 2022a) or Boolean features indicating if two nodes are in the same substructure (Bodnar et al., 2021). Examples of graph-level global structural encodings include eigenvalues of the adjacency or Laplacian (Kreuzer et al., 2021), or graph properties such as diameter, number of connected components, or treewidth.

On the other hand, positional encodings make, e.g., a node aware of its relative position to the other nodes in a graph. Hence, two such encodings should be close to each other if the corresponding nodes are close in the graph. Again, we can distinguish between local, global, or relative encodings. Examples of node-level local positional encodings include the shortest-path distance of a node to a hub or central node or the sum of each column of the non-diagonal elements of the $m$-step random walk matrix. Examples of edge-level relative positional encodings are pair-wise node distances (Beaini et al., 2021; Chen et al., 2022a; Kreuzer et al., 2021; Mialon et al., 2021; Li et al., 2020) and relative random walk encodings (Ma et al., 2023). Examples of node-level global positional encodings include eigenvectors of the graph Laplacian (Kreuzer et al., 2021; Dwivedi & Bresson, 2020) (or of the Magnetic Laplacian in case of directed graphs (Geisler et al., 2023)), or unique identifiers for each connected component of the graph. Lim et al. (2022) propose SignNet and BasisNet, two positional encodings also based on the eigenvectors of the graph Laplacian, which generalize a number of previously introduced structural and positional encodings such as those based on random walks (Dwivedi & Bresson, 2020; Mialon et al., 2021) or PageRank (Li et al., 2020).

When designing such encodings, one must ensure equivariance or invariance to the nodes' ordering. Such equivariance is trivially satisfied for simple encodings such as node degree but not for more powerful encodings

based on eigenvectors of the adjacency or Laplacian matrix (Lim et al., 2022). It is an ongoing effort to design equivariant Laplacian-based encodings (Lim et al., 2022; Wang et al., 2022).

## 2.3 Input Features

Besides characterizing GTs based on their use of structural and positional encodings, we can also characterize them based on their ability to deal with different node and edge features. To this end, we devise two families of input features. First, we consider so-called *non-geometric features* where nodes and edges have feature vectors in $\mathbb{R}^d$, i.e., graphs are described with a tuple $(V, E, \mathbf{X}, \mathbf{E})$. Secondly, we consider so-called *geometric features* where nodes and edges features contain geometric information, e.g., 3D coordinates for nodes $\mathbf{X}^{3D} \in \mathbb{R}^3$. Therefore, graphs are described with $(V, E, \mathbf{X}, \mathbf{E}, \mathbf{X}^{3D}, \mathbf{E}^{3D})$. We categorize GT architectures as non-geometric and those supporting both features in the following.

Non-geometric GTs (Chen et al., 2022b; Choromanski et al., 2021; Dwivedi & Bresson, 2020; He et al., 2022; Kim et al., 2022; Kreuzer et al., 2021; Jain et al., 2021; Ma et al., 2023; Mialon et al., 2021; Rampášek et al., 2022) are most common and follow the equations in Section 1.1. Graphs with non-geometric features do not have explicit geometric inductive bias. Examples of such features include encoded node attributes in citation networks or learnable atom-type embeddings in molecular graphs. Non-geometric features are supposed to be *equivariant* to node permutations, and transformers provide such equivariance by default. Structural and positional features (Section 2.2) are often added to increase the expressive power of GTs.

3D molecular graphs provide geometric features describing nodes and edges, e.g., 3D coordinates of atoms, angles of bonds, or torsion angles of planes. Building GTs supporting geometric features is more challenging as geometric features need to be *invariant* or *equivariant* to certain group transformations, such as rotation, depending on the task. Further, the architectures must be invariant for graph-level molecular property prediction tasks. In contrast, models must be equivariant in node-level tasks such as predicting structural conformers or force fields.

Joshi et al. (2023) showed that equivariant geometric models are more expressive than invariant ones. Here, we first describe $SO(3)$, $SE(3)$, and $E(3)$ equivariant models and then turn the attention to invariant models. TorchMD-NET (Thölke & Fabritiis, 2022) achieves $SO(3)$ equivariance by incorporating interatomic distances into the attention operation via exponential normal radial basis functions (RBF). SE(3)-Transformer (Fuchs et al., 2020) was one of the first attempts to incorporate $SE(3)$ equivariance. By using irreducible representations, Clebsch-Gordan coefficients, and spherical harmonics, the authors encode $SE(3)$ equivariance into the attention operation. Equiformer (Liao & Smidt, 2023) further extends this mechanism to complete $E(3)$ equivariance. Graphormer (Shi et al., 2022), Transformer-M (Luo et al., 2022a) and GPS++ (Masters et al., 2022) use Gaussian kernels to encode 3D distances between all pairs of atoms. Tailored for graph-level prediction tasks, GPS++ remains SE(3)-invariant, while Graphormer and Transformer-M introduce an additional SE(3)-equivariant prediction head for node-level molecular dynamics tasks.

## 2.4 Graph to Sequence Tokenization

The nature of *graph tokenization*, i.e., mapping a graph into a sequence of *tokens*, directly affects the supported features and computational complexity. Here, we identify three approaches to graph tokenization: (1) nodes as tokens, (2) nodes and edges as tokens, and (3) patches or subgraphs as tokens.

Using nodes as input tokens is the most common approach followed by many GTs, e.g., (Dwivedi & Bresson, 2020; Fuchs et al., 2020; Kreuzer et al., 2021; Luo et al., 2022b; Rampášek et al., 2022; Thölke & Fabritiis, 2022; Ying et al., 2021). Here, we often treat structural and positional features as additional node features. Given a graph with $n$ nodes and the attention procedure of Eq. (1), the complexity of such GTs is in $\mathcal{O}(n^2)$. We note that more scalable, sparse attention mechanisms are also possible; see Section 2.5. Edge features, e.g., shortest-path distances (Ying et al., 2021), random walk relative distances (Ma et al., 2023), or relative 3D distances (Luo et al., 2022b; Thölke & Fabritiis, 2022), may be added as an attention bias given the fully computed attention score matrix with $n^2$ entries. Alternatively, Mialon et al. (2021); Jain et al. (2021); Chen et al. (2022b) leverage a GNN to incorporate node and edge features before applying a transformer on the resulting node features. However, the transformer's quadratic complexity remains the bottleneck.

The second approach uses nodes and edges in the input sequence as employed by EGT (Hussain et al., 2022) and TokenGT (Kim et al., 2022). Turning an input graph into a graph of its edges is often used in molecular GNNs (Gasteiger et al., 2021) and NLP (Yao et al., 2020). In addition to soft modeling the edges, i.e., the *node-to-node* interactions, the attention operation also possibly models higher-order *node-edge* and *edge-edge* interactions that theoretically result in an expressiveness boost (Kim et al., 2022). The input sequence can naturally incorporate node features, their positional encodings, and edge features. A pitfall of this approach is its $\mathcal{O}(n+m)^2$ computational complexity. Since the approach includes edge features in the input sequence, such GTs might benefit from sparse attention mechanisms that do not materialize the full attention matrix.

The third approach relies on *patches* or *subgraphs* as tokens. In visual transformers (Dosovitskiy et al., 2021), such patches correspond to $k \times k$ image slices. A generalization of patches to the graph domain often corresponds to graph coarsening or partitioning (Baek et al., 2021; Chen et al., 2023; Kuang et al., 2022; He et al., 2022). There, tokens are small subgraphs extracted with various strategies. Initial representations of tokens are obtained by passing subgraphs through a GNN using a form of pooling to a single vector. He et al. (2022) adds token position features to the resulting vectors to distinguish coarsened subgraphs better. Finally, these tokens are passed through a transformer with $\mathcal{O}(k^2)$ complexity for a graph with $k$ extracted subgraphs. A similar approach is taken by NAGphormer (Chen et al., 2023) where tokens denote aggregated $l$-hop neighborhoods of a node. Each node is represented with a sequence of up to $L$ tokens including the node itself and each token for $l \in [1, L-1]$-hop neighborhood. Subgraph tokenization is more scalable since the model never sees the whole graph and might benefit from sampling methods. On the other hand, it makes GTs inherently local and might hinder their long-range capabilities.

## 2.5 Message Propagation

Most GTs follow the global all-to-all attention of Vaswani et al. (2017) between all pairs of tokens. In the initial GT (Dwivedi & Bresson, 2020) and TokenGT (Kim et al., 2022) this mechanism is unchanged, relying on token representations augmented with graph structural or positional information. Other models alter the global attention mechanism to bias it explicitly, typically based on the input graph's structural properties. Graphormer (Ying et al., 2021) incorporates shortest-path distances, representation of edges along a shortest path, and node degrees. Transformer-M (Luo et al., 2022a) follows Graphormer and adds kernelized 3D inter-atomic distances. GRPE (Park et al., 2022) considers multiplicative interactions of keys and queries with node and edge features instead of Graphormer's additive bias and additionally augments output token values. SAN (Kreuzer et al., 2021) relies on positional encodings and only adds preferential bias to interactions along input-graph edges over long-distance virtual edges. GraphiT (Mialon et al., 2021) employs diffusion kernel bias, while SAT (Chen et al., 2022b) develops a GNN-based structure-aware attention kernel. EGT (Hussain et al., 2022) includes a mechanism akin to cross-attention to edge tokens to bias inter-node attention and update edge representations. Finally, Zhang et al. (2023) propose Graphormer-GD, a generalization of Graphormer, alongside new expressivity metrics based on graph biconnectivity for which Graphormer-GD attains the necessary expressivity.

As standard global attention incurs quadratic computational complexity, it limits the application of graph transformers to graphs of up to several thousands of nodes. To alleviate this scaling issue, Choromanski et al. (2022) proposed GKAT based on a kernelized attention mechanism of the Performer (Choromanski et al., 2021), scaling linearly with the number of tokens. Another approach to improve GTs' scaling is to consider a reduced attention scope, e.g., based on locality or sparsified instead of dense all-to-all, following expander graph-based propagation (Deac et al., 2022) as in Exphormer (Shirzad et al., 2023). Finally, Wu et al. (2022) propose to view attention as in Equation (3) and then apply the kernel trick to $k_{\exp}$ to derive NodeFormer with a computational complexity that scales linearly with the number of nodes.

Finally, hybrid approaches combine several propagation schemes. For example, GPS and GPS++ (Rampášek et al., 2022; Masters et al., 2022) fuse local GNN-like architectures with global all-to-all attention into one layer. While GPS employs standard attention and can utilize linear attention mechanisms such as Performer (Choromanski et al., 2022), GPS++ follows Transformer-M's attention conditioning. GraphTrans (Jain et al., 2021) is also a hybrid but applies a stack of GNN layers first, followed by a stack of global transformer layers. Specformer (Bo et al., 2023) employs transformer encoder layers for encoding eigenvalue representations and mixes those with node features in the graph convolution-style decoder. GOAT (Kong et al., 2023) lets the nodes attend to $k$ cluster

centroids instead of $n$ nodes on the global attention level. The centroids are obtained through $k$-means and exponential moving average. On the local level, nodes attend to their $k$-hop neighborhood.

## 3    Applications of Graph Transformers

Although GTs only emerged recently, they have already been applied in various application areas, most notably in molecular property prediction. In the following, we give an overview of the applications of GTs. Kan et al. (2022) propose the Brain Network Transformers to predict properties of brain networks, e.g., the presence of diseases stemming from magnetic resonance imaging. To that, they leverage rows of the adjacency matrix of each node as structural encodings, which showed superior performance over Laplacian-based encodings in previous studies. Moreover, they devise a custom pooling layer leveraging the fact that nodes in the same functional module tend to have similar properties. Liao & Smidt (2023); Thölke & Fabritiis (2022), see also Section 2.3, devise an equivariant transformer architecture to predict quantum mechanical properties of molecules. To capture the molecular structure, they encode atom types and the atomic neighborhood into a vectorial representation, followed by a multi-head attention mechanism. To predict scalar atom-wise prediction, they rely on gated equivariant blocks (Schütt et al., 2021), which are then aggregated into single molecular predictions. Hu et al. (2020) develop an approach to apply graph transformers to web-scale heterogeneous graphs. During the attention computation, the authors use different projection matrices for each node and edge relation in the heterogeneous graph. Yao et al. (2020) use transformers to tackle the graph-to-sequence problem, i.e., the problem of translating a graph to word sequences. They first translate a graph to its Levi graph, replacing labeled edges with additional nodes to incorporate edge labels. They then split such a graph into multiple subgraphs according to the different edge nodes. Each subgraph uses a standard transformer architecture to learn the vectorial representation for each node. To incorporate graph structure, they mask out non-neighbors of a node, concentrating on the local structure. Finally, they concatenate multiple node representations. Further applications use transformers for rumor detection in microblogs (Khoo et al., 2020), predicting properties of crystals (Yan et al., 2022) or click-through rates (Min et al., 2022b), or leverage them for 3D human pose and mesh reconstruction from a single image (Lin et al., 2021a).

## 4    Experimental Study

Here, we conduct an empirical study to complement our taxonomy in a separate direction. Concretely, we empirically evaluate two highly discussed aspects of graph transformers: (1) the effectiveness of incorporating *graph structural bias* into GTs, and (2) their ability to reduce *over-smoothing* and *over-squashing*. This study aims to compare selected methods from our taxonomy and investigate graph transformers more generally in terms of their potential benefits over GNNs. Note that an empirical evaluation of all presented works would be out of scope for the present work. Instead we focus on the approaches most prevalent in the literature. Concretely, we aim to answer the following questions.

**Q1** How well do different strategies for incorporating structural awareness into GTs contribute to recovering fundamental structural properties of graphs?

**Q2** Does the ability of transformers to reduce over-smoothing lead to improved performance on heterophilic datasets?

**Q3** Do graph transformers alleviate over-squashing better than GNN models?

### 4.1    Structural Awareness of GTs

For question **Q1**, we evaluate the two most prevalent strategies for incorporating graph structure bias into transformers.

**Positional and Structural Encodings** (Section 2.2). Random-walk structural encodings (RWSE) and Laplacian positional encodings (LapPE), two popular positional or structural encodings for transformers (Rampášek et al., 2022).

**Attention Bias** (Section 2.5). Attention bias based on spatial information such as shortest-path distance between nodes, following the Graphormer architecture (Ying et al., 2021).

Table 1: Hyper-parameter sets for GTs and GNNs with or without PE/SE (SET 1), and for Graphormer models (SET 2).

| Hyper-parameter | SET 1 | SET 2 |
|---|---|---|
| Embedding dim. | 64 | 72 |
| Self-attention heads | 4 | 4 |
| Weight decay | $10^{-5}$ | $10^{-2}$ |
| Learning rate | $10^{-3}$ | $10^{-3}$ |
| Gradient clip norm | 1.0 | 5.0 |
| LR scheduler | cosine, warm-up | constant |
| Batch size | 96 | 256 |

Table 2: Average test accuracy of GTs with structural bias ($\pm$ SD) over five random seeds on the structural awareness tasks. Difficulty level on top derived from GIN performance. We additionally report the performance of a transformer without any structural bias serving as a baseline.

| Model | Easy
EDGES | Medium
TRIANGLES-SMALL | Medium
TRIANGLES-LARGE | Hard
CSL |
|---|---|---|---|---|
| | 2-way Accuracy $\uparrow$ | 10-way Accuracy $\uparrow$ | 10-way Accuracy $\uparrow$ | 10-way Accuracy $\uparrow$ |
| GIN | $98.11_{\pm1.78}$ | $71.53_{\pm0.94}$ | $33.54_{\pm0.30}$ | $10.00_{\pm0.00}$ |
| Transformer | $55.84_{\pm0.32}$ | $12.08_{\pm0.31}$ | $10.01_{\pm0.04}$ | $10.00_{\pm0.00}$ |
| Transformer (LapPE) | $98.00_{\pm1.03}$ | $78.29_{\pm0.25}$ | $10.64_{\pm2.94}$ | $100.00_{\pm0.00}$ |
| Transformer (RWSE) | $97.11_{\pm1.73}$ | $99.40_{\pm0.10}$ | $54.76_{\pm7.24}$ | $100.00_{\pm0.00}$ |
| Graphormer | $97.67_{\pm0.97}$ | $99.09_{\pm0.31}$ | $42.34_{\pm6.48}$ | $90.00_{\pm0.00}$ |

We propose a benchmark of three tasks that require increasingly higher structural awareness of non-geometric graphs. We determine the level of structural awareness necessary to solve a task according to the baseline performance of GIN (Xu et al., 2019), a 1-WL expressive GNN reference architecture. In addition, we report the performance of a vanilla transformer without any structural bias to understand the relative impact of the positional or structural encodings (PE/SE) and attention biasing.

We first describe the tasks in our benchmark and their estimated difficulty, then outline task-specific hyper-parameters of evaluated models and interpret the observed results; see Table 2 for quantitative results.

**Detect Edges (Easy).** Detecting whether an edge connects two nodes can be considered the fundamental test for structural awareness. We investigate this task using a custom dataset, EDGES, derived from the ZINC (Dwivedi et al., 2023) dataset. For each graph, we treat the pairs of nodes connected by an edge as positive examples and select an equal number of unconnected nodes as negative examples, resulting in a binary edge detection task with balanced classes. Let $P$ denote the set of pairs selected as either positive or negative examples, and let $\mathbf{h}_v^{(T)}$ denote the feature vector of node $v$ after the last layer $T$ of a model. We make predictions as follows. We first compute the cosine similarity between $\mathbf{h}_v^{(T)}$ and $\mathbf{h}_w^{(T)}$ for each pair $(v, w)$ of nodes in $P$, resulting in a scalar similarity score. Finally, we apply a linear layer to each similarity score, followed by a sigmoid activation, resulting in binary class probabilities.

**Count Triangles (Medium).** Counting triangles only requires information within a node's immediate neighborhood. However, more than 1-WL expressivity is required to solve it (Morris et al., 2019). Hence, a standard GIN architecture is not powerful enough. For this task, we evaluate models on the TRIANGLES dataset proposed by Knyazev et al. (2019), which poses triangle counting as a 10-way classification problem. Here, graphs have between 1 and 10 triangles, each corresponding to one class. The dataset specifies a fixed train/validation/test split, which we adopt in our experiments. Graphs in the train and validation split are roughly the same size. The test set is a mixture of two graph distributions, where 50% are graphs with a similar size to those in the training and validation set (TRIANGLES-SMALL) and 50% are graphs of larger size (TRIANGLES-LARGE). We separately report model

performance for TRIANGLES-SMALL and TRIANGLES-LARGE to study the ability of transformers with different structural biases to generalize to larger graphs. We analyzed the datasets' class balance and report that each test set contains 5000 graphs with 500 graphs per class. For more details about the dataset, see Knyazev et al. (2019).

**Distinguish Circular Skip Links (CSL) (Hard).** A 1-WL limited model cannot distinguish non-isomorphic CSL graphs (Murphy et al., 2019) as the task requires an understanding of distance (Morris et al., 2019). Here, we evaluate models on the CSL dataset (Dwivedi et al., 2023), which contains 150 graphs with skip-link lengths ranging from 2 to 16 and poses a 10-way classification problem. We follow Dwivedi et al. (2023) in training with 5-fold cross-validation.

**Hyper-parameters.** To simplify hyper-parameter selection, we hand-designed two general sets of hyper-parameters; see Table 1. For EDGES and TRIANGLES, we fix a parameter budget of around 200k for the transformer models, resulting in six layers for each model with the respective embedding sizes specified in Table 1. Further, we train Graphormer on 1k epochs. Due to the small number of graphs in the CSL dataset, we fix a parameter budget of around 100k for the transformer models, resulting in three layers for each model with the exact embedding sizes as above. Further, we train Graphormer on 2k epochs. We repeat each experiment on five random seeds and report the model accuracy's mean and standard deviation.

For our 1-WL-equivalent reference model, we chose the GIN layer (Xu et al., 2019). To improve comparability with the transformer models, we use a feed-forward neural network composed of the same components, using the same hyper-parameters as for transformers. For Graphormer, we use the feed-forward neural network specified by Ying et al. (2021). Further, we train GIN with the SET 1 hyper-parameters. For EDGES and TRIANGLES, this results in around 150k parameters, while for CSL, the GIN model contains approximately 75k parameters.

**Answering Q1.** Table 2 shows that GTs supplemented with structural bias generally perform well on all three tasks with a few exceptions. First, the GT with Laplacian positional encodings performs sub-par on the TRIANGLES task. However, it is still an improvement over the 1-WL-equivalent GIN. We hypothesize this is due to an expressivity limit of Laplacian encodings regarding triangle counting. Secondly, we observe that all models generalize poorly to larger graphs on the TRIANGLES dataset. Lastly, we observe that on CSL, Graphormer cannot surpass 90% accuracy. A deeper analysis revealed that the shortest-path distributions can only distinguish 9 out of the 10 classes correctly, meaning that Graphormer is theoretically limited to at most 90% accuracy on CSL.

The above failure cases highlight that current graph transformers still suffer from limited expressivity, and no clear expressivity hierarchy exists for the used positional or structural encodings. Moreover, GTs may generalize poorly to larger graphs. At the same time, we demonstrate a general superiority of structurally biased GTs over standard 1-WL-equivalent models such as GIN. Both the transformer with RWSE as well as Graphormer solve EDGES, TRIANGLES-SMALL, and CSL almost perfectly, two of which pose a challenge for GIN, especially on CSL where GIN performs no better than random.

### 4.2 Reduced Over-smoothing in GTs?

Graph transformers are often ascribed with an ability to circumvent GNNs' over-smoothing problem due to their global attention mechanism. Thus, we set out to benchmark several variants of GCN (Kipf & Welling, 2017), hybrid GPS models, and Graphormer on six heterophilic transductive datasets: ACTOR (Tang et al., 2009); CORNELL, TEXAS, WISCONSIN (CMU, 2001); CHAMELEON and SQUIRREL (Rozemberczki et al., 2021). In addition, we also consider the recently proposed heterophilic datasets in (Platonov et al., 2023), which are significantly larger (ranging from 10k to 45k nodes), where we benchmark GCN (Kipf & Welling, 2017), GAT (Veličković et al., 2018), as well as GPS. While over-smoothing can occur both on homophilic and heterophilic graphs, only on heterophilic graphs is over-smoothing truly limiting. This is because successfully predicting a nodes' class on a heterophilic graph potentially requires a model to take into account nodes further away than those in the intermediate one or two-hop neighborhood and hence requires locally aggregating GNNs to be sufficiently deep. However, since graph transformers allow interactions between all pairs of nodes fewer layers might suffice to successfully solve a heterophilic problem.

For the six small datasets, we broadly follow the SET 1 hyper-parameters (Table 1). However, we perform a grid search for each model variant to select the embedding size (32, 64, or 96) and dropout rates while we fix the

Table 3: Benchmarking of multiple model variants on six heterophilic transductive datasets. Here we report average test accuracy (± SD) over ten random seeds. We follow the dataset protocol of Pei et al. (2020); for additional model comparison; see Luan et al. (2022).

| Model (PE/SE type) | ACTOR | CORNELL | TEXAS | WISCONSIN | CHAMELEON | SQUIRREL |
|---|---|---|---|---|---|---|
| Geom-GCN (Pei et al., 2020) | 31.59 ±1.15 | 60.54 ±3.67 | 64.51 ±3.66 | 66.76 ±2.72 | **60.00** ±2.81 | 38.15 ±0.92 |
| GCN (no PE/SE) | 33.92 ±0.63 | 53.78 ±3.07 | 65.95 ±3.67 | 66.67 ±2.63 | 43.14 ±1.33 | 30.70 ±1.17 |
| GCN (LapPE) | 34.30 ±1.12 | 56.22 ±2.65 | 65.95 ±3.67 | 66.47 ±1.37 | 43.53 ±1.45 | 30.80 ±1.38 |
| GCN (RWSE) | 33.69 ±1.07 | 53.78 ±4.09 | 62.97 ±3.21 | 69.41 ±2.66 | 43.84 ±1.68 | 31.77 ±0.65 |
| GCN (DEG) | 33.99 ±0.91 | 53.51 ±2.65 | 66.76 ±2.72 | 67.26 ±1.53 | 46.36 ±2.07 | 34.50 ±0.87 |
| GPS$^{\text{GCN+Transformer}}$ (LapPE) | 37.68 ±0.52 | 66.22 ±3.87 | 75.41 ±1.46 | 74.71 ±2.97 | 48.57 ±1.02 | 35.58 ±0.58 |
| GPS$^{\text{GCN+Transformer}}$ (RWSE) | 36.95 ±0.65 | 65.14 ±5.73 | 73.51 ±2.65 | **78.04** ±2.88 | 47.57 ±0.90 | 34.78 ±1.21 |
| GPS$^{\text{GCN+Transformer}}$ (DEG) | 36.91 ±0.56 | 64.05 ±2.43 | 73.51 ±3.59 | 75.49 ±4.23 | 52.59 ±1.81 | 42.24 ±1.09 |
| Transformer (LapPE) | **38.43** ±0.87 | **69.46** ±1.73 | **77.84** ±1.08 | 76.08 ±1.92 | 49.69 ±1.11 | 35.77 ±0.50 |
| Transformer (RWSE) | **38.13** ±0.63 | **70.81** ±2.02 | **77.57** ±1.24 | **80.20** ±2.23 | 49.45 ±1.34 | 35.35 ±0.75 |
| Transformer (DEG) | 37.39 ±0.50 | **71.89** ±2.48 | **77.30** ±1.32 | **79.80** ±0.90 | 56.18 ±0.83 | **43.64** ±0.65 |
| Graphormer (DEG only) | 36.91 ±0.85 | 68.38 ±1.73 | 76.76 ±1.79 | 77.06 ±1.97 | 54.08 ±2.35 | **43.20** ±0.82 |
| Graphormer (DEG, attn. bias) | 36.69 ±0.70 | 68.38 ±1.73 | 76.22 ±2.36 | 77.65 ±2.00 | 53.84 ±2.32 | **43.75** ±0.59 |

Table 4: Benchmarking of multiple model variants on the five transductive node classification datasets proposed in (Platonov et al., 2023). Here we report average test accuracy (± SD) for ROMAN-EMPIRE and AMAZON-RATINGS and ROC AUC (± SD) for MINESWEEPER, TOLOKERS and QUESTIONS. Results over 10 splits, following (Platonov et al., 2023). We highlight the **first**, **second** and **third** best results.

| Model (PE/SE type) | ROMAN-EMPIRE | AMAZON-RATINGS | MINESWEEPER | TOLOKERS | QUESTIONS |
|---|---|---|---|---|---|
| GCN Platonov et al. (2023) | 73.69 ±0.74 | 48.70 ±0.63 | 89.75 ±0.52 | 83.64 ±0.67 | 76.09 ±1.27 |
| GAT Platonov et al. (2023) | 80.87 ±0.30 | 49.09 ±0.63 | 92.01 ±0.68 | 83.70 ±0.47 | 77.43 ±1.20 |
| GCN (LapPE) | 83.37 ±0.55 | 44.35 ±0.36 | **94.26** ±0.49 | 84.95 ±0.78 | **77.79** ±1.34 |
| GCN (RWSE) | 84.84 ±0.55 | 46.40 ±0.55 | **93.84** ±0.48 | **85.11** ±0.77 | **77.81** ±1.40 |
| GCN (DEG) | 84.21 ±0.47 | **50.01** ±0.69 | 94.14 ±0.50 | 82.51 ±0.83 | 76.96 ±1.21 |
| GAT (LapPE) | 84.80 ±0.46 | 44.90 ±0.73 | 93.50 ±0.54 | **84.99** ±0.54 | 76.55 ±0.84 |
| GAT (RWSE) | **86.62** ±0.53 | 48.58 ±0.41 | 92.53 ±0.65 | **85.02** ±0.67 | 77.83 ±1.22 |
| GAT (DEG) | 85.51 ±0.56 | **51.65** ±0.60 | 93.04 ±0.62 | 84.22 ±0.81 | 77.10 ±1.23 |
| GPS$^{\text{GCN+Performer}}$ (LapPE) | 83.96 ±0.53 | 48.20 ±0.67 | **93.85** ±0.41 | 84.72 ±0.77 | **77.85** ±1.25 |
| GPS$^{\text{GCN+Performer}}$ (RWSE) | 84.72 ±0.65 | 48.08 ±0.85 | 92.88 ±0.50 | 84.81 ±0.86 | 76.45 ±1.51 |
| GPS$^{\text{GCN+Performer}}$ (DEG) | 83.38 ±0.68 | 48.93 ±0.47 | 93.60 ±0.47 | 80.49 ±0.97 | 74.24 ±1.18 |
| GPS$^{\text{GAT+Performer}}$ (LapPE) | **85.93** ±0.52 | 48.86 ±0.38 | 92.62 ±0.79 | 84.62 ±0.54 | 76.71 ±0.98 |
| GPS$^{\text{GAT+Performer}}$ (RWSE) | **87.04** ±0.58 | 49.92 ±0.68 | 91.08 ±0.58 | 84.38 ±0.91 | 77.14 ±1.49 |
| GPS$^{\text{GAT+Performer}}$ (DEG) | 85.54 ±0.58 | **51.03** ±0.60 | 91.52 ±0.46 | 82.45 ±0.89 | 76.51 ±1.19 |
| GPS$^{\text{GCN+Transformer}}$ (LapPE) | OOM | OOM | 91.82 ±0.41 | 83.51 ±0.93 | OOM |
| GPS$^{\text{GCN+Transformer}}$ (RWSE) | OOM | OOM | 91.17 ±0.51 | 83.53 ±1.06 | OOM |
| GPS$^{\text{GCN+Transformer}}$ (DEG) | OOM | OOM | 91.76 ±0.61 | 80.82 ±0.95 | OOM |
| GPS$^{\text{GAT+Transformer}}$ (LapPE) | OOM | OOM | 92.29 ±0.61 | 84.70 ±0.56 | OOM |
| GPS$^{\text{GAT+Transformer}}$ (RWSE) | OOM | OOM | 90.82 ±0.56 | 84.01 ±0.96 | OOM |
| GPS$^{\text{GAT+Transformer}}$ (DEG) | OOM | OOM | 91.58 ±0.56 | 81.89 ±0.85 | OOM |

number of layers to two. We implement the GCN and GT models following GPS with hybrid GCN+Transformer aggregation layers but with the latter or former component disabled, respectively. We train all models in full-batch mode using the entire graph as input.

For the five large datasets, we closely follow the hyper-parameter tuning and training setup described in (Platonov et al., 2023). To this end, we set the embedding size to 512 and only tune the number of layers (1, 2, 3, 4, and 5). Because on these datasets, the number of nodes provides an additional challenge for the transformer, we benchmark GPS also with the Performer module (Choromanski et al., 2022) to study the benefits of linear attention on large-scale graphs.

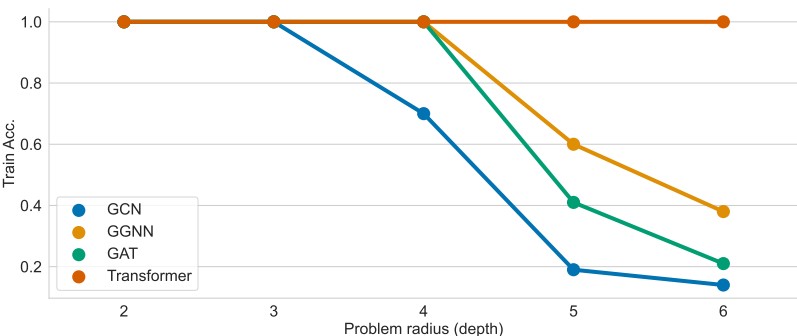

Figure 2: Average train accuracy over ten random seeds of a GT on the NEIGHBORSMATCH dataset, compared to models from Alon & Yahav (2021).

Apart from Graphormer, we benchmark all models on three different positional/structural encodings, namely LapPE, RWSE, and degree encodings as described in (Ying et al., 2021).

**Answering Q2.** On the small datasets (Table 3), all transformer-based models outperform a 2-layer GCN, and often the specialized Geom-GCN (Pei et al., 2020), which experimental setup we follow. With the exemption of node degree encodings (DEG), other PE/SE had minimal effect on GCN's performance. Adding global attention to the GCN, i.e., following the GPS model, universally improves the performance. Most interestingly, disabling the local GCN in GPS, i.e., becoming the Transformer model, increases the performance even further. Such results indicate that GNN-like models are unfit for these heterophilic datasets, while the global attention of a transformer empirically facilitates considerably more successful information propagation. Graphormer, which utilizes node degree encodings, performs comparably to the Transformer counterparts. Surprisingly, the attention bias of Graphormer had no impact on its performance. The shortest-path distance bias appears uninformative in these datasets, unlike, e.g., in ZINC, where we observed degradation from 0.12 test MAE to 0.54 when disabling the attention bias.

We conclude that we empirically confirm the expected benefits of global attention, albeit GTs do not achieve overall SOTA performance (e.g., see Luan et al. (2022)), which is a reminder that specialized architectures can achieve similar or higher performance without global attention still.

On the large datasets (Table 4), we observe strong benefits of added positional/structural encodings but also that their effect is highly dataset dependent. At the same time, an added Transformer or Performer module does not yield improvements in general. Specifically, the full Transformer goes out of memory on the three largest datasets while leading to poor performance on MINESWEEPER and competitive performance on TOLOKERS. Interestingly, the largest improvements of positional/structural encodings and Transformer modules are achieved on ROMAN-EMPIRE, the only dataset with a large graph diameter (6824).

Here, we conclude that positional/structural information can lead to large gains in performance, while global attention only provides small or no improvements at all. One possible explanation is that when applied to graphs with 10K or more nodes, the self-attention of graph transformers is subjected to much more noise, which makes learning meaningful long-range interactions difficult.

### 4.3 Reduced Over-squashing in GTs?

To answer question **Q3**, we evaluate a GT on the NEIGHBORSMATCH problem proposed by Alon & Yahav (2021). This synthetic dataset requires long-range interaction between leaf nodes and the root node of a tree graph of depth $d$. The problem demonstrates GNNs' limited ability to transmit information across a receptive field that grows exponentially with $d$. We run our experiments with minimal changes to the code of Alon & Yahav (2021) and train our transformer on depths 2 to 6. Note that GNN models fail to perfectly fit the training set of trees with depth 4. Convergence on NEIGHBORSMATCH can sometimes take up to 100k epochs for large depths $d$. Since the structure of the graphs in NEIGHBORSMATCH is irrelevant to solving the problem, we did not need to augment

node features with positional/structural encodings or attention bias. Hence, the results represent most graph transformers that use global aggregation similar to the standard transformer. Otherwise, we used the same architecture as in Section 4.1.

**Answering Q3.** Section 4.3 shows that the GT performs exceedingly better than the GNN baselines and can perfectly fit the training set even for depth $d = 6$. However, we note that the NEIGHBORSMATCH problem is idealized and has only limited practical implications. The core issue of over-squashing, which is squashing an exponentially growing amount of information into a fixed-size vector representation, is not resolved by transformers. Nonetheless, our results demonstrate that the ability of transformers to model long-range interactions between nodes can circumvent the problem posed by Alon & Yahav (2021).

## 5  Open Challenges and Future Work

Since the area of GTs is a new, emerging field, there are still many open challenges, both from practical and theoretical points of view. Theoretically, although it is often claimed that GTs offer better predictive performance over GNNs and are more capable of capturing long-range dependencies and preventing over-smoothing and over-squashing, a principled explanation still needs to be formed. Moreover, there needs to be a clearer understanding of the properties of structural and positional encodings. For example, it has yet to be understood when certain encodings are helpful and how they compare. The first step could be precisely characterizing different encodings to distinguish non-isomorphic graphs, similar to the Weisfeiler–Leman hierarchy (Morris et al., 2021). Further, understanding GTs generalization performance on larger graphs has yet to be understood similarly to GNNs (Yehudai et al., 2021; Zhou et al., 2022).

On the practical side, one major downside of GTs is their quadratic running time in the number of tokens, preventing them from scaling to truly large graphs typical in real-world node-level prediction tasks. Moreover, due to the attention mechanism's nature, how best to incorporate edge features into GT architectures still needs to be determined. Further, our experimental analysis reveals a disadvantage of local GNN-like models when used in conjunction with transformers as in Rampášek et al. (2022) on heterophilic datasets. Heterophily is thus an open challenge, also for GTs. Moreover, it is crucial to incorporate expert or domain knowledge, e.g., physical or chemical knowledge for molecular prediction, into the attention matrix in a principled manner.

Explaining and interpreting the performance of GTs remains an open research area where we draw parallels with NLP. We posit that studying GT in the graph ML community follows a similar path of studying transformer language models in NLP unified under the name of *Bertology* (Rogers et al., 2021; Vulić et al., 2020). Numerous Bertology studies reported that more than dissecting attention matrices and attention scores in transformer layers is needed for understanding how language models work. The community converged on designing datasets and tasks tailored to language model features such as co-reference resolution or mathematical reasoning. Therefore, we hypothesize that understanding GTs' performance through attention scores is limited, and the community should focus on designing diverse benchmarks probing particular GTs' properties. Further, studying scaling laws and emergent behavior of GTs and GNNs is still in its infancy, with few examples in chemistry (Frey et al., 2022) and protein representation learning (Lin et al., 2022).

## 6  Conclusion

We have provided a taxonomy of graph transformers (GTs). To this end, we overviewed GTs' theoretical properties and their connection to structural and positional encodings. We then thoroughly surveyed how GTs can deal with different input features, e.g., 3D information, and discussed the different ways of mapping a graph to a sequence of tokens serving as GTs' input. Moreover, we thoroughly discussed different ways GTs propagate information and recent real-world applications. Most importantly, we showed empirically that different encodings and architectural choices drastically impact GTs' predictive performance. We verified that GTs can deal with heterophilic graphs and prevent over-squashing to some extent. Finally, we proposed open challenges and outlined future work. We hope our survey presents a helpful handbook of graph transformers' methods, perspectives, and limitations and that its insights and principles will help spur and shape novel research results in this emerging field.

## Acknowledgements

CM and LM are partially funded by a DFG Emmy Noether grant (468502433) and RWTH JPI Fellowship under Germany's Excellence Strategy.

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

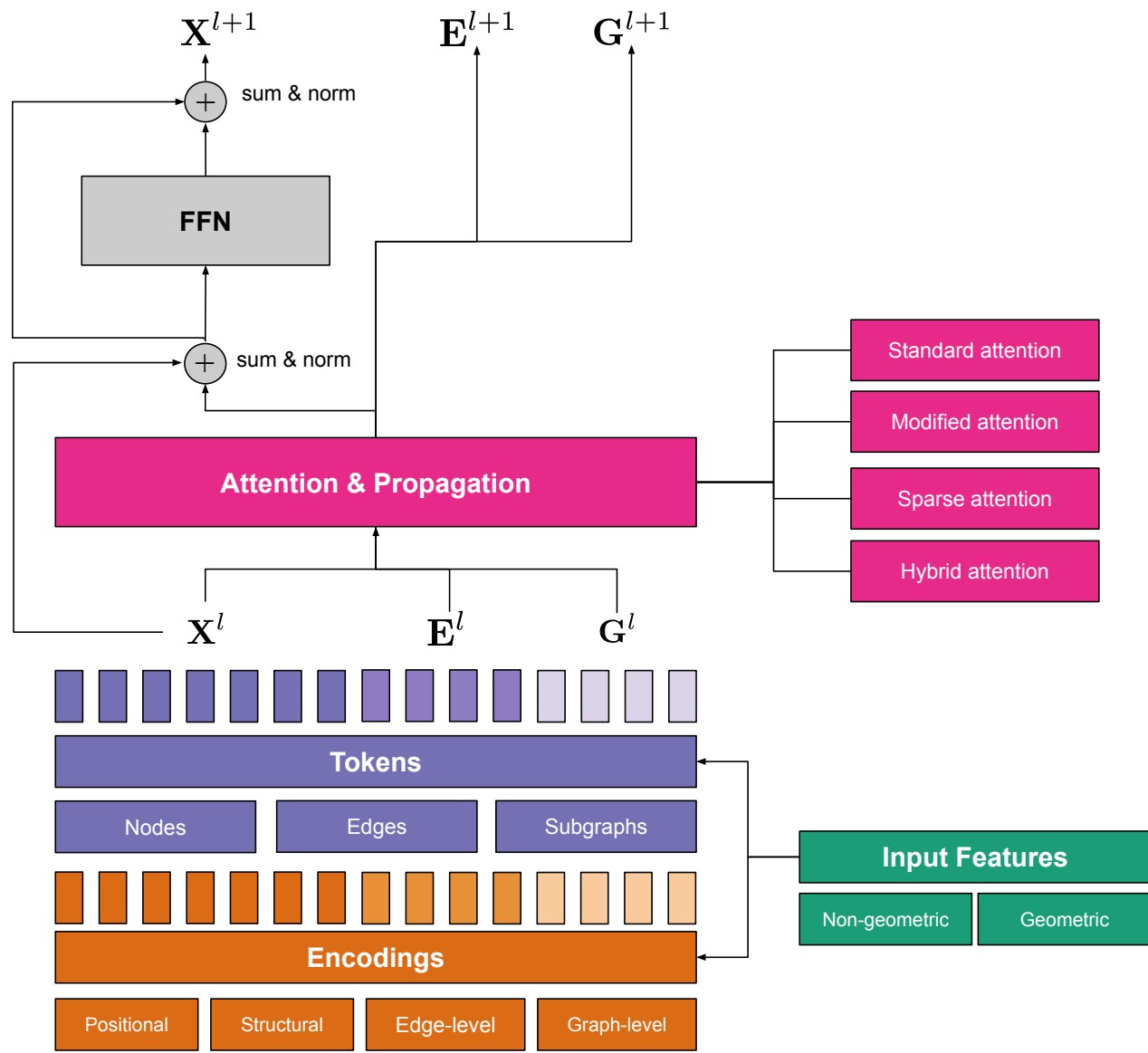

Figure 3: Overview of how the different branches of our taxonomy affect the original transformer architecture.

## A    Dataset Details

Here, we describe the datasets used in our experiments; see Table 5 for an overview over the dataset statistics.

**Edges**   We derive the EDGES dataset from ZINC (Dwivedi et al., 2023). ZINC comprises 12K molecules from the ZINC database with heavy atoms as nodes and bonds as edges. The original task of ZINC is to predict the constrained solubility of a given molecule. Note that we ignore these regression targets in our link prediction task.

**Triangles**   The TRIANGLES dataset contains synthetic graphs with the goal of predicting the number of triangle subgraphs contained in a given graph (Knyazev et al., 2019).

**CSL**   The CSL dataset contains synthetic regular graphs, i.e., graphs where each node has the same number of neighbors (Dwivedi et al., 2023). The nodes of each graph initially form a cycle and are then additionally equipped

with additional edges, so-called skip links, connecting nodes on the cycle that are some fixed length $L$ apart. Different graphs are constructed for different skip link lengths $L$ and the task of CSL is to classify $L$.

**Actor** The ACTOR dataset is derived from the film-director-actor-writer network, a heterogeneous graph resulting from crawled Wikipedia pages and proposed in Tang et al. (2009). Specifically, ACTOR is a graph where actors are the nodes, and two actors are connected by an edge if their names occur on the same Wikipedia page. The task is to classify the actors in one of the five most occurring categories in the category information on the respective actor's Wikipedia page.

**Cornell, Texas, Wisconsin** The CORNELL, TEXAS and WISCONSIN are three of the four webpage networks collected in WebKB (CMU, 2001). Each network contains crawled webpages as nodes and hyperlinks connecting the pages as edges. The task is to predict which category of student, project, course, staff, and faculty a webpage belongs to.

**Chameleon, Squirrel** The CHAMELEON and SQUIRREL are datasets constructed from Wikipedia pages of their respective topic (that of Chameleons and that of Squirrels, respectively), where the nodes are articles and edges represent links from one article to another (Rozemberczki et al., 2021). The node features are based on the occurrence of particular nouns on the respective article and the task is to predict the average monthly traffic on the site.

**Roman-Empire** The ROMAN-EMPIRE dataset is based on the Wikipedia article of the Roman Empire (Platonov et al., 2023), where nodes are the words occurring in the article and an edge between two words indicates that either the words follow each other in a sentence or if one word syntactically depends on the other. The task is to predict the syntactic role of the work according to spaCy (Honnibal et al., 2020).

**Amazon-Ratings** The AMAZON-RATINGS dataset is derived from Amazon product co-purchasing metadata (Platonov et al., 2023), where the nodes are products and an edge between two products indicates that the products are frequently bought together. The task is to predict the average rating of the product from one to five stars.

**Minesweeper** The MINESWEEPER dataset is constructed from a grid graph, where each node represents a cell in a Minesweeper game, and the task is to predict for each cell whether it contains a mine (Platonov et al., 2023).

**Tolokers** The TOLOKERS dataset contains data from the Toloka crowd-sourcing platform (Platonov et al., 2023). Nodes represent workers, and an edge between two workers indicates that the workers have worked together on one of 13 selected projects. The task is to predict which workers have been banned from a project.

**Questions** The QUESTIONS dataset contains data from a question-answering wbesite (Platonov et al., 2023). Nodes represent users answering questions on medicine, and an edge between two users indicates that the users have answered the same questions within some fixed time span. The task is to predict which users remained active at the end of the time span.

## B  Experimental details

Here, we describe the details of our experiments. All experiments were run on a single A100 NVIDIA GPU with 80GB of GPU RAM.

We base our implementation on GraphGPS (Rampášek et al., 2022), which is available at `https://github.com/rampasek/GraphGPS`, which also includes an implementation of Graphormer (Ying et al., 2021), except for the over-squashing experiment, where we use the implementation by Alon & Yahav (2021) to stay as close as possible to the original implementation. All model layers first apply a convolution/attention, followed by a feed-forward network. The resulting embeddings are then fed into a final MLP head to make a prediction. For all models, we use a GELU non-linearity (Hendrycks & Gimpel, 2016) in the feed-forward network. Further, we apply residual connections for all models, one after the convolution/attention and one after the feed-forward network.

Table 5: Statistics of the datasets used in our experiments (Dwivedi et al., 2023; Knyazev et al., 2019; Tang et al., 2009; CMU, 2001; Rozemberczki et al., 2021; Platonov et al., 2023)

| Dataset | Num. graphs | Num. nodes | Num. edges | Task | Metric |
|---|---|---|---|---|---|
| EDGES | 12,000 | 277,864 | 597,970 | Link prediction | Cross entropy |
| TRIANGLES | 45,000 | 938,438 | 2,947,024 | 10-way graph classification | Cross entropy |
| CSL | 150 | 6,150 | 24,600 | 10-way graph classification | Cross entropy |
| ACTOR | 1 | 7,600 | 30,019 | 5-way node classification | Cross entropy |
| CORNELL | 1 | 183 | 298 | 5-way node classification | Cross entropy |
| TEXAS | 1 | 183 | 325 | 5-way node classification | Cross entropy |
| WISCONSIN | 1 | 251 | 515 | 5-way node classification | Cross entropy |
| CHAMELEON | 1 | 2,277 | 36,101 | 5-way node classification | Cross entropy |
| SQUIRREL | 1 | 5,201 | 217,073 | 5-way node classification | Cross entropy |
| ROMAN-EMPIRE | 1 | 22,662 | 32,927 | 18-way node classification | Cross entropy |
| AMAZON-RATINGS | 1 | 24,492 | 93,050 | 5-way node classification | Cross entropy |
| MINESWEEPER | 1 | 10,000 | 39,402 | 2-way node classification | Cross entropy |
| TOLOKERS | 1 | 11,758 | 519,000 | 2-way node classification | Cross entropy |
| QUESTIONS | 1 | 48,921 | 153,540 | 2-way node classification | Cross entropy |

For the structural awareness experiments in Section 4.1, we follow the hyper-parameters detailed in Table 1. In addition, for all models, we use six layers of convolution/attention. For the GIN, Transformer, and GPS, we use batch normalization, mean pooling, and three layers for the final MLP head. For Graphormer, we use layer normalization, `[graph]` token readout, and a linear layer, preceded by a final layer norm for the final MLP head, following (Ying et al., 2021).

For the six small datasets ACTOR, CORNELL, TEXAS, WISCONSIN, CHAMELEON and SQUIRREL, we select hyper-parameters with a grid search over the hidden dimension (32, 64, 96), dropout (0.0, 0.2, 0.5, 0.8) and, where applicable, attention dropout (0.0, 0.2, 0.5). For the grid search, we repeat each experiment ten times and select the hyper-parameters leading to the best average validation performance. In all models, we use two layers of convolution/attention and a linear layer for the final MLP head. We use batch normalization for the GCN, GPS, and Transformer and layer normalization for Graphormer. Again, for Graphormer, we apply a final layer norm before the final MLP head.

For the five large datasets in Platonov et al. (2023), we follow their hyper-parameter selection exactly to enable a fair comparison. As a result, we only tune the number of layers (1,2,3,4,5). We use sum pooling and a linear layer for the final MLP head.

