# OpenReview forum: "Attending to Graph Transformers"
_TMLR — Accepted by TMLR_

### Review · Reviewer_7Lg5 · 2023-11-29

**Summary Of Contributions:**

This paper serves as a comprehensive survey, focusing on recent developments in Graph Transformer (GT) architectures. It specifically provides a taxonomy of Graph Transformers, offering an organized framework for understanding these models. The authors delve into the theoretical properties of Graph Transformers and their relationship with structural and positional encodings. This survey also covers the diverse ways in which Graph Transformers process various input features.

Furthermore, they discuss the different methods through which Graph Transformers disseminate information, alongside their recent applications in real-world scenarios. Additionally, their investigation extends to empirical analysis, examining Graph Transformers' ability to discern various graph properties, handle heterophilic graphs, and mitigate the over-squashing phenomenon. Lastly, the paper summarizes the current challenges in the field of Graph Transformers and proposes directions for future research, paving the way for further advancements in this domain.

**Audience:**

Yes

**Broader Impact Concerns:**

I do not have any concerns about the ethical implications of the work.

**Claims And Evidence:**

Yes

**Requested Changes:**

I would like to suggest that the authors enhance the paper by including illustrative figures for some of the more influential methods, in order to deepen the technical understanding. Additionally, incorporating a table that summarizes the datasets, along with the specific tasks they are intended for and their estimated difficulty levels, would greatly improve the paper's clarity and usefulness.

**Strengths And Weaknesses:**

Strengths:
1. Comprehensive Coverage: The survey provides an extensive and timely overview of various Graph Transformer methods. Its broad scope ensures a thorough understanding of the field's current state and recent developments.
2. Clarity and Accessibility: The paper is particularly commendable for its clear and accessible presentation, especially regarding the motivation and taxonomy of Graph Transformers. This approach makes it easier for newcomers to quickly grasp the essential concepts of the graph transformer field.
3. Empirical Contribution: Empirical Contribution: Beyond merely summarizing existing methods, the survey incorporates empirical studies to validate previous concepts. This addition is a valuable contribution to the community.

Weakness:
The primary limitation of this survey is its limited depth, particularly for those new to the field. While it effectively categorizes GT methods from various angles, such as Structural and Positional Encodings, Message Propagation, and Graph to Sequence Tokenization, the paper relies heavily on text descriptions when detailing model differences. To enhance understanding, it would be beneficial to incorporate illustrative figures that highlight key trends and distinctions. Additionally, for the datasets featured in the survey, a tabular summary outlining the specific tasks they are designed for and their estimated difficulty levels would be valuable. This approach would provide readers with a clearer and more comprehensive overview of the datasets and their relevance in the context of graph transformers.

---

> ### Author Response · Authors · 2024-01-13
> **Response to reviewer 7Lg5**
>
> We thank the reviewer for their constructive review and their suggestions. We have updated the manuscript accordingly. First, we have included a comprehensive figure in the appendix that demonstrates how the different branches of the taxonomy are reflected in architectural changes to the original transformer. Second, we updated the appendix to describe each task presented in the empirical study in more detail.

---

### Review · Reviewer_Jn7n · 2023-12-19

**Summary Of Contributions:**

The current paper is a technical survey of graph transformer methodologies and includes a theoretical analysis of the models, empirical differences between them, and the potential of future extensions.

The survey starts with a small background overview of graph theory, graph transformers, and graph neural networks. Subsequently, the literature is broken down and analyzed in detail based on five different criteria. The first is the models’ provable expressive power and overall theoretical properties. The second is the encoding type of the input features, while the third is the input features category.
The fourth section refers to approaches that utilize the graph as a sequence similar to NLP. The final section separates papers based on their way of using attention to combine neighboring nodes’ features.
The subsequent sections include a short overview of applications and an experimental study that aims to examine the ability of graph transformers to recover structural properties, perform on heterophilic datasets, and reduce over-squashing. The extensive experiments indicate a ranking between a number of graph transformer methods in all three tasks. The answers are informative and concise.

Finally, the paper concludes with a summary of current limitations and potential future work.
Overall, the paper is interpretable and inclusive enough to be considered a useful handbook for graph transformers.

**Audience:**

Yes

**Broader Impact Concerns:**

There is no evident concern as the paper is a survey.

**Claims And Evidence:**

Yes

**Requested Changes:**

Apart from addressing the aforementioned weaknesses, some structural properties of the paper such as:
Figure 1 should be closer to section 2 (possibly page 4). The hierarchical structure could include more information to make the scheme more independent or the information could be added in the caption.

All tables in the experimental section should be closer to the text describing them.

I believe it is more sensible to first introduce GNNs and then move to graph transformers, and the same applies to sections 2.2 and 2.3.

**Strengths And Weaknesses:**

Strengths
The survey is in general quite inclusive and timely.

The experiments are useful as they indeed provide sensible answers.


Weaknesses:
The criteria of the taxonomy are disconnected from the experiments. The paper is a set of useful sections but is not connected sufficiently. It could be interesting to expand upon how the experimental answers correlate with the taxonomy e.g. how does over-smoothing correlate with the type of input features and message passing? It is important for a survey to be consistent.

The theoretical considerations, which are a vital part of separating the current survey from others, are quite limited. The authors address briefly the models’ expressive power and some comparison with GNNs, but do not include differentiation of the methods in Figure 1 based on their theoretical properties neither they expand enough to understand the intuition behind some arguments of the paper e.g. why SE(3) group defines a clear distinction in 2.3.

The applications could be more elaborate, the list misses for example this prominent paper:
Hu, Z., Dong, Y., Wang, K., & Sun, Y. (2020, April). Heterogeneous graph transformer. In Proceedings of the Web Conference 2020 (pp. 2704-271

---

> ### Author Response · Authors · 2024-01-13
> **Response to reviewer Jn7n**
>
> We thank the reviewer for their constructive review and address their remarks in the following.
>
> > It could be interesting to expand upon how the experimental answers correlate with the taxonomy e.g. how does over-smoothing correlate with the type of input features and message passing?
>
> **Answer**: We want to point out that our Q2 studies exactly this question. If the predictive performance on heterophilic graphs is indeed an indicator of how well models deal with over-smoothing, our results in Tables 3 and 4 show how different types of input features (Laplacian eigenvectors and -values, random-walk information, degree information) and different types of message-passing fare in the heterophilic setting (GPS+GCN vs. Graphormer in Table 3 and GPS+GCN vs. GPS+GAT in Table 4).
>
> > [authors] do not include differentiation of the methods in Figure 1 based on their theoretical properties
>
> **Answer**: Here, we would like to ask the reviewer for clarification about whether they want Figure 1 to additionally reflect the theoretical properties of the methods in a separate (sub-) branch or whether the reviewer wants us to expand in the text on the theoretical properties of these methods. In both cases, understanding theoretical properties such as the expressivity of graph transformers is an ongoing research problem and would escape the scope of our work.
>
> > [authors do not] expand enough to understand the intuition behind some arguments of the paper e.g. why SE(3) group defines a clear distinction in 2.3
>
> **Answer**: We have updated the manuscript to highlight this distinction more clearly.
>
> > The applications could be more elaborate, the list misses for example this prominent paper [...]
>
> **Answer**: We agree that this paper should be included and updated the manuscript accordingly.
>
> > All tables in the experimental section should be closer to the text describing them.
>
> **Answer**: We agreed and moved all tables and figures closer to the corresponding text.
>
> > I believe it is more sensible to first introduce GNNs and then move to graph transformers, and the same applies to sections 2.2 and 2.3
>
> **Answer**: Since this work focuses on transformers for graphs, introducing GTs first was a deliberate choice aimed at viewing the field of graph learning through the lens of GTs rather than the other way around.

---

> > ### Comment · Reviewer_Jn7n · 2024-01-14
> > **Response to authors**
> >
> > I thank the authors for their answers and their updates in the manuscript.
> >
> > > We want to point out that our Q2 studies exactly this question...
> >
> > Indeed measuring the AUC in heterophilic tasks only does not suffice to facilitate the comparison, as over-smoothing is a problem in both, heterophillic and homophillic graphs, and the latter are equally if not more prevalent. If the survey focuses exclusively on heterophilic graphs, it should be mentioned in the title, although it severely restricts the scope. If not, a set of experiments on some of the well-known homophilic graphs (of varying sizes) could improve the benchmark study significantly.
> >
> > > Here, we would like to ask the reviewer for clarification..
> >
> > I was referring to the former i.e. given that the survey does have theoretical considerations, I would expect the figure to convey some of the text accordingly.
> >
> > >  We agree that this paper should be included..
> >
> > Since it is a seminal work, it would make sense to add it as an extra benchmark or clarify why you limit your choice of transformers in these models.

---

> > > ### Author Response · Authors · 2024-01-18
> > > **Response to reviewer**
> > >
> > > We appreciate the detailed discussion about our work.
> > >
> > > > Indeed measuring the AUC in heterophilic tasks only does not suffice to facilitate the comparison [...]
> > >
> > > **Answer**: It is true that over-smoothing can occur both on homophilic and heterophilic graphs. However, only on heterophilic graphs is over-smoothing truly limiting since, successfully predicting a nodes’ class on a heterophilic graph potentially requires a model to take into account nodes further away than those in the intermediate one or two-hop neighborhood and hence requires locally aggregating GNNs to be sufficiently deep. However, since graph transformers allow interactions between all pairs of nodes fewer layers might suffice to successfully solve a heterophilic problem. As a result, we study shallow models with two layers and demonstrate that graph transformers perform indeed significantly better than locally aggregating GNNs on heterophilic graphs, indicating that the global aggregation is able to account for the lack of depth in the GT.
> > >
> > > > Since it is a seminal work, it would make sense to add it as an extra benchmark or clarify why you limit your choice of transformers in these models.
> > >
> > > To our understanding, the Heterogeneous Graph Transformer is specifically developed for heterogeneous graphs. However, our study is merely focused on homogeneous graphs (in the heterophilic settings) and as such the HGT, while surely relevant to the application section of our survey, is out of scope for our experimental study.

---

> > > > ### Comment · Reviewer_Jn7n · 2024-01-18
> > > > **Response to authors**
> > > >
> > > > > It is true that over-smoothing can occur both on homophilic..
> > > >
> > > > I understand the intuition of your argument. That said, it should be clarified in the document, e.g. with explicit references and theoretical/experimental proofs, such that the survey's conclusions do not generalize to homophilic graphs.

---

> > > > > ### Author Response · Authors · 2024-01-21
> > > > > **Response to reviewer**
> > > > >
> > > > > We updated the manuscript to include our explanation of why the experimental study is limited to heterophilic graphs.

---

### Review · Reviewer_Limv · 2023-12-20

**Summary Of Contributions:**

The paper presents a comprehensive study of graph transformer architectures in machine learning. It introduces a taxonomy of graph transformers which categorizes graph transformers based on several key aspects. It includes the types of transformers, focusing on their architecture and how they process graph data. The taxonomy considers aspects like how these transformers encode structural and positional information of graphs, and how they handle different input features. This classification helps in understanding the diverse approaches and methods used in graph transformer models, providing a structured way to compare and analyze their capabilities and applications in various machine learning tasks. The paper also explores their applications, particularly in 3D molecular graphs, and investigates their effectiveness in recovering graph properties, dealing with heterophilic graphs, and preventing over-squashing. The paper aims to provide insights and directions for future research in this field.

**Audience:**

Yes

**Claims And Evidence:**

No

**Requested Changes:**

I think the paper will be a very valuable contribution. However, the experiment section as it stands today is very lacking. A paper proposing to give taxonomy for graph transformer should provide a detailed study across the various axes in the taxonomy (at least to a degree that seems doable). Only one question (Q1) largely compared along a proposed axis in the taxonomy. However, the other two question (Q2 and Q3) largely dealt with GNNs vs GT questions. I would suggest incorporating more questions that compares GTs along more axes which will likely provide a detailed understanding of GTs while proving the value of the proposed taxonomy for research purposes.

**Strengths And Weaknesses:**

# Strengths
1. The paper has comprehensive evaluation of graph transformers.
2. It offers a detailed taxonomy, providing clarity on different types of graph transformer architectures.

# Weaknesses
1. The taxonomy raises several questions for study - how do various encoding strategies fare for various downstream tasks?, which transformers scale better with graph size?, etc. However, one of these questions have been answered in the experiment section.
2. The experiment section largely deals with comparing GNN and GTs on oversmoothing and oversquashing problems with some discussion on comparison with GTs. While this is valuable result to know, but it is unclear as to how the insights provided connect with the taxonomy proposed in the paper.

---

> ### Author Response · Authors · 2024-01-13
> **Response to reviewer Limv**
>
> We thank the reviewer for their constructive review and address their remarks in the following.
>
> > A paper proposing to give taxonomy for graph transformer should provide a detailed study across the various axes in the taxonomy (at least to a degree that seems doable)
>
> **Answer**: Our paper does not claim to provide a complete empirical study comparing our taxonomy's different branches. Rather, our empirical study complements the taxonomy by providing additional insights about the most prominent graph transformers (and positional/structural encodings) introduced in the taxonomy. We have updated the manuscript to highlight this more clearly. Our claims include the taxonomy and survey of existing graph transformer models, probing representative GTs in several tasks, and outlining open challenges. With that, we believe the manuscript's claims and content are aligned.
>
> > The taxonomy raises several questions for study - how do various encoding strategies fare for various downstream tasks?, which transformers scale better with graph size?, etc. However, one of these questions have been answered in the experiment section.
>
> **Answer**: We point out that our study indeed addresses these questions. First, regarding a comparison of encoding strategies for various downstream tasks, in Table 2, Table 3, and Table 4, we compare various encoding strategies, namely Laplacian encodings, random-walk encodings, degree encodings and the Graphormer encodings. Secondly, regarding the transformer scaling, the graphs in Table 4 are sufficiently large to bring the full attention out of memory while Performer-attention successfully scales to these graphs and performs better than full attention on the two graphs where full attention can be applied. This indicates that graph transformers such as GPS that can use the Performer present a viable strategy for very large graphs.

---

### Decision · Action_Editor_eo6f · 2024-01-31

**Recommendation:** Accept as is

**Comment:**

The chief concern among reviewers was that the two main contributions of this work (the taxonomy and the empirical results) appear disconnected. For example, the empirical study raises questions about "over-smoothing" and "over-squashing" (Q2, Q3) but these are not discussed at all in Section 2, which concerns the taxonomy. Conversely, many of the differences between methods included in the taxonomy are not evaluated in the empirical study.

These discrepancies do not make the paper incorrect, or take away from potential interest, but I fear that many readers will have the same reaction to the paper, even after the first revision made by the authors. For the camera-ready version, I recommend that the presentation be improved further to align or compare the two goals of the paper. For example, does the taxonomy help in answering the empirical questions or is it entirely separate? Why is the selected subset of methods used in the empirical study appropriate, in light of the larger taxonomy?

**Audience:**

Yes. The survey is timely and could be of large interest.

**Claims And Evidence:**

The evidence corresponds well with the stated contributions (Present work in the Introduction).